# Antijamming Improvement for Frequency Hopping Using Noise-Jammer Power Estimator

**Hojun Lee** **, Jongmin Ahn, Yongcheol Kim and Jaehak Chung** *

Electronics engineering, Inha University, 100, Inha-ro, Michuhol-gu, Incheon 22212, Korea;
timmit@naver.com (H.L.); anjong3@naver.com (J.A.); dydcjf4691@naver.com (Y.K.)
* Correspondence: jchung@inha.ac.kr; Tel.: +82-032-874-7421

**Abstract:** In frequency-hopping spread-spectrum (FHSS) systems, jammer detection and mitigation are important but difficult. Each slot of the FHSS experiences frequency-selective fading and unequal transceiver-frequency gains that hinder the detection of jammed slots and result in a poor bit-error rate (BER). To increase BER performance, we first propose a noise-jammer power estimator (NJPE) that estimates noise and jammer powers regardless of different channel gains, and derived its normalized Cramér–Rao bound (NCRB). Second, we developed a jammer detector based on gamma distribution, and designed a restoration method combining all nonjammed slots. Computer simulations verified the derived NCRB of the proposed NJPE by normalized mean squared error (NMSE), and showed that the jammer-detection probability of the proposed jammer detector was better than that of conventional detectors. The BER performance of the proposed method was also shown to be better than that of conventional methods.

**Keywords:** jammer; military communication; jammer detection; jammer mitigation; frequency-hopping spread spectrum; Cramér–Rao bound

## 1. Introduction

In military wireless communication systems, the bit-error rate (BER) in a receiver is degraded when a transmitted signal is jammed [1,2]. Military wireless communications need to mitigate jammers to attain reliability. To reduce jamming, several antijam techniques, such as frequency-hopping spread spectrum (FHSS) and direct-sequence spread spectrum (DSSS), were researched [3–15]. When jammer power is large, the FHSS is known to be more effective [16], but BER degradation remains.

FHSS randomly changes carrier frequency according to time within a transmission-frequency band, which reduces jammer probability for a slot. As a jammer to FHSS, a partial-band noise jammer (PBNJ) was considered in this study [17,18]. When jamming occurs, the BER of jammed slots is so large that the receiver may not be able to decode the transmitted information. To overcome the jammer problem, a duplication method was studied where several slots with the same data are transmitted and combined to increase signal-to-jammer-plus-noise ratio (SJNR) [19]. However, when the jammer was strong, the SJNR was still small. A selective combining method was developed [19], and the detection of jammed slot methods was also researched [20,21]. In FHSS systems, however, multipath channels cause frequency-selective fading, and nonlinear devices at the transmitter also generate an unequal frequency response. Since the unequal frequency response makes the detection of jammed slots difficult, a solution for measuring jammer power on all empty slots and detecting jammed slots was developed [21]. This full search of all empty slots requires large computational complexity.

To overcome jammer-detection and -mitigation problems, we developed a proposal for a noise-jammer power estimator (NJPE), a jammer detector based on gamma distribution, and a slot combiner for nonjammed slots. The proposed NJPE estimates the pure power of the noise and

jammer of each slot regardless of transmitted-symbol existence and unequal frequency response. The probability distribution of the NJPE output was approximated as gamma distribution on the basis of simulations, and the jammer detector based on gamma distribution was developed to precisely detect the jammer. The normalized Cramér–Rao bound (NCRB) of the proposed NJPE was also derived to theoretically prove estimation performance. Computer simulations demonstrated that the normalized mean squared error (NMSE) of the proposed NJPE was well-matched with the derived NCRB, and the detection performance of the jammer detector based on gamma distribution was better than those of conventional jammer detectors. The BER of the proposed method indicated better performance than that of conventional methods.

This paper consists of five sections: Section 2 discusses how the proposed NJPE works; Section 3 analyzes how to assess the performance of the proposed NJPE; Section 4 compares the NCRB of the proposed scheme with NMSE and BER performance using computer simulations; and Section 5 concludes the paper.

## 2. Proposed Estimator and Detector

In FHSS systems for military communications, the same data are repeatedly transmitted over several slots and combined in the receiver to mitigate jammers. When jammer power is strong, and jammed slots are not discarded, BER performance degrades. Thus, the detection of jammed slots is important. In FHSS systems, however, frequency-selective fading and unequal frequency responses cause different slot powers and make the detection of jammed slots difficult. In this section, we show the developed NJPE and the jammer detector based on gamma distribution of NJPE. The proposed detector can detect both narrowband and wideband jammers. Figure 1 demonstrates the proposed method that detects and discards jammed slots, and combines the remaining slots.

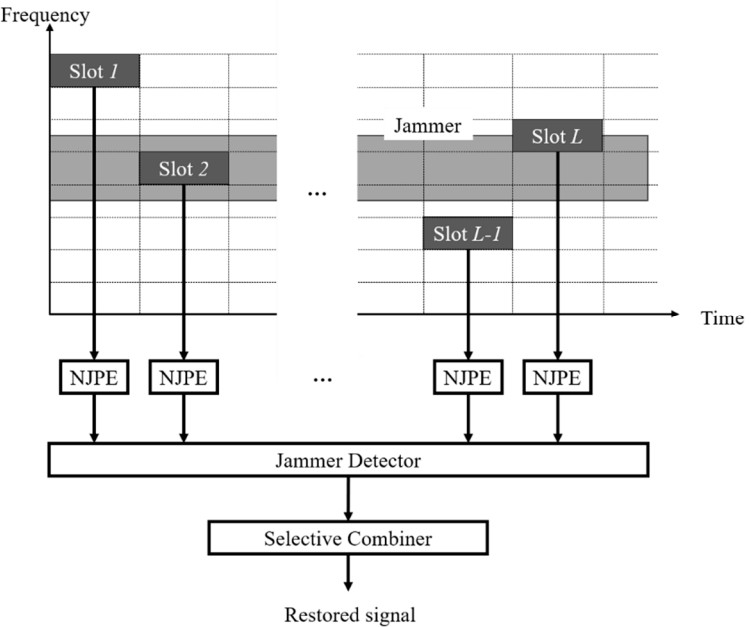

**Figure 1.** Block diagram of proposed method.

*2.1. Noise-Jammer Power Estimator*

In order to estimate pure noise-jammer power regardless of transmitted-symbol power, the received symbol was multiplied by its conjugation, and its mean and variance were calculated. With a simple manipulation of mean and variance, the pure noise-jammer power was easily estimated.

Assume that the data of a slot are copied and repeatedly transmitted to $L$ slots, the received symbol of the $l$-th slot is $r_l$, and $X_l$ is the multiplication of $r_l$ and its conjugation $r_l^*$. Then, $X_l$ can be written as

$$
\begin{aligned}
X_l &= r_l r_l^* \\
&= (h_l s_l + \widetilde{n}_l)(h_l s_l + \widetilde{n}_l)^* \\
&= |h_l|^2 |s_l|^2 + \widetilde{n}_l s_l^* h_l^* + h_l s_l \widetilde{n}_l^* + \left|\widetilde{n}_l\right|^2,
\end{aligned}
\tag{1}
$$

where $h_l$ denotes a channel coefficient of the $l$-th slot, $s_l$ denotes a transmitted symbol of the $l$-th slot, and * denotes a complex conjugate. $h_l$ is uncorrelated to other channel coefficients by frequency-selective fading. $\widetilde{n}_l$ denotes noise with or without jammer on the $l$-th slot, which is defined by

$$
\widetilde{n}_l = \begin{cases} n_l, & \text{w.o. jammer} \\ n_l + j_l, & \text{w. jammer} \end{cases},
\tag{2}
$$

where $n_l$ and $j_l$ denote pure noise and the jammer on the $l$-th slot, respectively.

To develop the proposed NJPE, the mean and variance of $X_l$ in Equation (1) were calculated. Assume that noise, transmitted symbol, and channel gain are independent. The PBNJ was modeled as zero-mean Gaussian noise [22,23]. Therefore, the mean of $X_l$ was obtained as

$$
\begin{aligned}
\mathrm{E}[X_l] &= \mathrm{E}\left[|h_l|^2|s_l|^2\right] + \mathrm{E}[\widetilde{n}_l s_l^* h_l^*] + \mathrm{E}[h_l s_l \widetilde{n}_l^*] + \mathrm{E}\left[\left|\widetilde{n}_l\right|^2\right] \\
&= \begin{cases} \mathrm{E}_s \mathrm{E}\left[|h_l|^2\right] + \sigma_{l,N}{}^2, & \text{w.o. jammer} \\ \mathrm{E}_s \mathrm{E}\left[|h_l|^2\right] + \sigma_{l,N}{}^2 + \sigma_{l,J}{}^2, & \text{w. jammer} \end{cases},
\end{aligned}
\tag{3}
$$

where $\sigma_{l,N}{}^2$ and $\sigma_{l,J}{}^2$ denote the pure noise power and jammer power on the $l$-th slot, respectively, and $\mathrm{E}_s$ denotes a symbol power. Next, the variance of $X_l$ was obtained as

$$
\begin{aligned}
\mathrm{Var}[X_l] = \quad & \mathrm{Var}\left[|h_l|^2|s_l|^2\right] + \mathrm{Var}[\widetilde{n}_l s_l^* h_l^*] + \mathrm{Var}[h_l s_l \widetilde{n}_l^*] + \mathrm{Var}\left[\left|\widetilde{n}_l\right|^2\right] + 2\mathrm{Cov}\left(|h_l|^2|s_l|^2, \widetilde{n}_l s_l^* h_l^*\right) \\
& + 2\mathrm{Cov}\left(|h_l|^2|s_l|^2, h_l s_l \widetilde{n}_l^*\right) + 2\mathrm{Cov}\left(|h_l|^2|s_l|^2, \left|\widetilde{n}_l\right|^2\right) + 2\mathrm{Cov}\left(\widetilde{n}_l s_l^* h_l^*, h_l s_l \widetilde{n}_l^*\right) \\
& + 2\mathrm{Cov}\left(\widetilde{n}_l s_l^* h_l^*, \left|\widetilde{n}_l\right|^2\right) + 2\mathrm{Cov}\left(h_l s_l \widetilde{n}_l^*, \left|\widetilde{n}_l\right|^2\right).
\end{aligned}
\tag{4}
$$

Equation (4) was classified as with- or without-jammer since $\widetilde{n}_l$ had with- or without-jammer cases in Equation (2). In FHSS, the length of each slot was designed to be shorter than coherence time, and channel gain was assumed as constant in a slot. Then, Equation (4) was simplified as

$$
\mathrm{Var}[X_l] = \begin{cases} \sigma_{l,N}{}^4 + 2\mathrm{E}_s\mathrm{E}\left[|h_l|^2\right]\sigma_{l,N}{}^2, & \text{w.o. jammer} \\ \left(\sigma_{l,N}{}^2 + \sigma_{l,J}{}^2\right)^2 + 2\mathrm{E}_s\mathrm{E}\left[|h_l|^2\right]\left(\sigma_{l,N}{}^2 + \sigma_{l,J}{}^2\right), & \text{w. jammer} \end{cases}.
\tag{5}
$$

Equations (3) and (5) both have noise, symbol, jammer powers, and channel gain. To estimate the pure noise-jammer power regardless of channel gain, $\mathrm{E}_s\mathrm{E}\left[|h_l|^2\right]$ in Equation (3) needed to be removed. $\mathrm{E}_s\mathrm{E}\left[|h_l|^2\right]$ was calculated by taking the square root of $\left(\mathrm{E}_s\mathrm{E}\left[|h_l|^2\right]\right)^2$, which was obtained by subtracting the squared Equation (3) from Equation (5). Then, the proposed NJPE could be derived as

$$
\begin{aligned}
Z_l &= \mathrm{E}[X_l] - \sqrt{\mathrm{E}[X_l]^2 - \mathrm{Var}[X_l]} \\
&= \begin{cases} \sigma_{l,N}{}^2, & \text{w.o. jammer} \\ \sigma_{l,N}{}^2 + \sigma_{l,J}{}^2, & \text{w. jammer} \end{cases}.
\end{aligned}
\tag{6}
$$

In Equation (6), if a jammer existed, NJPE estimated the sum of pure noise and jammer power; if not, NJPE only estimated pure noise power. Because the noise was generated from a low-noise amplifier (LNA) at the receiver, the noise powers of all slots were similar. Therefore, jammed slots

were easily detected by comparing with other slots. The estimation performance of the proposed NJPE is explained in Section 3.

## 2.2. Jammer Detector wirh Gamma Distribution

Jammer detectors determine a jammed slot when an estimated noise power of a slot is greater than those of the other slots. A robust jammer-detection method is the constant false-alarm-rate (CFAR) detector, which calculates a threshold obtained from a given false-alarm-rate probability ($P_{fa}$) and compares the threshold with a test sample. The conventional CFAR detector of radar systems assumes that background noise follows Gaussian distribution. However, the output distribution of the proposed NJPE was not Gaussian because of the square and square-root terms in Equation (6).

In this study, we approximated that the NJPE output followed gamma distribution on the basis of simulations, and derived the jammer detector based on gamma distribution. In the experiment section, we demonstrate that the detection performance of the derived jammer detector attained better detection probability than those of conventional Gaussian-distribution-based detectors.

To calculate a relationship between threshold and false-alarm rate $P_{fa}$, let an actual and an estimated noise-jammer power be $\theta$ and $z$, respectively. The conditional probability-density function (PDF) of $z$ given $\theta$ ($f_Z(z|\theta)$), follows gamma distribution and is given as (see Appendix A)

$$f_Z(z|\theta) = \frac{1}{\Gamma(\alpha(\theta))\beta(\theta)^{\alpha(\theta)}} z^{\alpha(\theta)-1} \exp(-z/\beta(\theta)), \tag{7}$$

where $\alpha(\theta)$ and $\beta(\theta)$ denote

$$\alpha(\theta) = \frac{NP_{r,l}^2}{\theta^2 + 4P_{r,l}\theta + 2P_{r,l}^2}, \tag{8}$$

$$\beta(\theta) = \frac{\theta(\theta^2 + 4P_{r,l}\theta + 2P_{r,l}^2)}{NP_{r,l}^2}, \tag{9}$$

where $N$ denotes the number of samples, and $P_{r,l}$ denotes a received symbol power of the $l$-th slot ($E_s E[|h_l|^2]$) from Equation (3). Then, a relationship between $P_{fa}$ and threshold $V_T$ of the proposed jammer detector is obtained as

$$P_{fa} = \int_{V_T}^{\infty} f_Z(z|\theta)dz = \frac{1}{\Gamma(\alpha(\theta))}\Gamma\left(\alpha(\theta), \frac{V_T}{\beta(\theta)}\right), \tag{10}$$

where $\Gamma(s, x)$ denotes an upper incomplete gamma function by

$$\Gamma(s, x) = \int_x^{\infty} t^{s-1}e^{-t}dt. \tag{11}$$

Thus, for a given $P_{fa}$, $V_T$ is calculated by Equation (10), and a jammed slot can be determined.

After detecting all jammed slots, the index set ($S_{comb}$) of the combining slots is obtained as

$$S_{comb} = \{l \mid Z_l < V_T, \ l \in \{1, 2, \ldots, L\}\}, \tag{12}$$

where $Z_l$ denotes the estimated power of the noise and jammer in the $l$-th slot from Equation (6). If the only nonjammed slots based on $S_{comb}$ were combined, a larger signal-to-noise ratio (SNR) and a better BER could be attained.

## 3. Proposed-Method Analysis

For NJPE, the limited number of symbols in a slot is utilized for calculation, which causes an estimation error. In this section, we derived the NCRB of the proposed NJPE, analyzed the detection performance of the proposed jammer detector, and calculated the detection probability.

The characteristic of the NCRB of the proposed NJPE is described in Lemma 1.

**Lemma 1.** *The estimation error of NJPE is inversely proportional to the number of symbols in a slot. When the SNR is large, the estimation error only depends on the number of symbols.*

**Proof.** Let the estimated noise power of the proposed NJPE be $\hat{\theta}$ for actual power $\theta$. The variance of $\hat{\theta}$ is bounded by Fisher information ($I(\theta)$), which is given as [24]

$$\mathrm{Var}\big[\hat{\theta}\big] \geq \frac{1}{I(\theta)}, \tag{13}$$

where $I(\theta)$ is calculated as (see Appendix B for detail derivation)

$$I(\theta) = -\mathrm{E}\left[\frac{\partial^2 \ln f_Z(z|\theta)}{\partial \theta^2}\right] = \frac{NP_r^2}{\theta^2(\theta^2 + 4P_r\theta + 2P_r^2)}. \tag{14}$$

where $N$ denotes the number of symbols, and $P_r$ denotes the received symbol power. Thus, the Cramér–Rao bound (CRB) is obtained as

$$\mathrm{Var}\big[\hat{\theta}\big] \geq CRB(\theta) = \frac{\theta^2\big(\theta^2 + 4P_r\theta + 2P_r^2\big)}{NP_r^2}. \tag{15}$$

NCRB is defined as $NCRB(\theta) \stackrel{\text{def}}{=} CRB(\theta)/\theta^2$ [25]. NCRB is written as

$$NCRB(\theta) = \frac{\theta^2 + 4P_r\theta + 2P_r^2}{NP_r^2} = \frac{1 + 4\Omega + 2\Omega^2}{N\Omega^2}, \tag{16}$$

where $\Omega = P_r/\theta$, which is SNR. In Equation (16), NCRB is inversely proportional to $N$ and is functional of $\Omega$. When $\Omega$ is large, NCRB becomes $2/N$. □

The detection probability ($P_D$) of the proposed jammer detector was calculated from threshold $V_T$, which was obtained from $P_{fa}$. Because the proposed NJPE followed gamma distribution, $P_D$ needed to be calculated on the basis of the conditional probability of the gamma function in Equation (7). If the jammer existed in the observation slot, $\theta$ in Equation (7) became the sum of the noise power and jammer power ($\sigma_N^2 + \sigma_J^2$). Then, $P_D$ with threshold $V_T$ was obtained by an integration over the region greater than $V_T$, which was given as

$$P_D = \int_{V_T}^{\infty} f_Z\big(z|\sigma_N^2 + \sigma_J^2\big)dz = \frac{1}{\Gamma\big(\alpha\big(\sigma_N^2 + \sigma_J^2\big)\big)}\Gamma\left(\alpha\big(\sigma_N^2 + \sigma_J^2\big), \frac{V_T}{\beta\big(\sigma_N^2 + \sigma_J^2\big)}\right), \tag{17}$$

where $\alpha\big(\sigma_N^2 + \sigma_J^2\big)$ and $\beta\big(\sigma_N^2 + \sigma_J^2\big)$ were defined as in Equations (8) and (9), respectively. Thus, in Equation (17), as $\alpha\big(\sigma_N^2 + \sigma_J^2\big)$ increased and $\beta\big(\sigma_N^2 + \sigma_J^2\big)$ decreased, $P_D$ increased. In other words, a larger SNR provided better detection probability. In the next section, we outline the computer simulations that were executed to show the detection performance of the proposed and conventional schemes.



## 4. Computer Simulations

The validation of gamma approximation and the derived NCRB were verified through computer simulations. The detection probability and the BER of the proposed methods were evaluated alongside conventional schemes. For the validation of gamma approximation, the distribution of the NJPE output was fitted to gamma, normal, and Weibull, and the root mean squared errors (RMSEs) of moments for the three fitted distributions were compared to select best-fit distribution. The derived NCRB was compared with the NMSE of the proposed NJPE. The detection probability of the proposed jammer detector based on gamma distribution was compared with those of the conventional cell average (CA)-CFAR, smallest of (SO)-CFAR, and differential jammer rejection (DJR) [20,21]. The BERs of the combining slots without jammed slots were tested alongside conventional methods.

Experiment parameters are shown in Table 1, where $\rho$ denotes the ratio of the bandwidths of FHSS system and jammer. A small $\rho$ indicates a strong and narrowband jammer for the same jammer power.

**Table 1.** Experiment parameters.

| Parameters | Values |
| --- | --- |
| Bandwidth | 1 MHz |
| Modulation scheme | $\pi/4$ differential QPSK |
| Repetition (the number of slots) | 4 |
| $E_b/N_0$ | 12 dB |
| $\rho$ | 0.0005, 0.005, 0.05 |
| Channel model | Rician channel |
| $P_{fa}$ | $10^{-10}$ |
| Number of symbols in a slot | 50 |

In order to verify gamma approximation, maximum-likelihood estimation was utilized to fit the distribution of the NJPE output to gamma, normal, and Weibull distributions [26], and the RMSEs of the $n$-th moment from 1 to 4 were compared to select the best-fit distribution.

The PDFs of the three fitted distributions and the RMSEs are shown in Figure 2 and Table 2, respectively. In Figure 2, the black solid line denotes the PDF of the output of NJPE. Red-triangle, green-square, and blue-circle lines denote the PDFs of the fitted distributions with gamma, normal, and Weibull, respectively. Gamma distribution was well-matched with the distribution of the NJPE output. In Table 2, $\mu_n$ denotes the $n$-th moment, and the gamma distribution displayed the smallest RMSEs from the three fitted distributions.

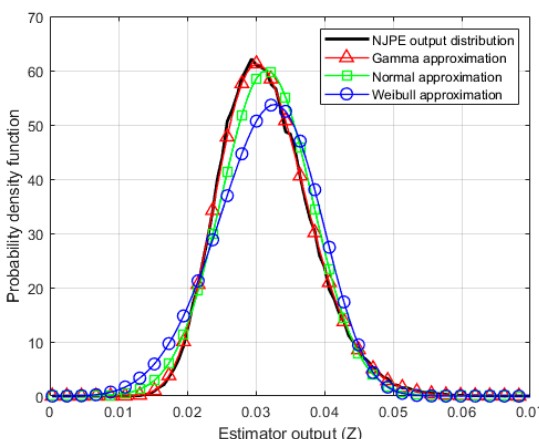

**Figure 2.** Probability-density functions of fitted distributions.

**Table 2.** Root mean squared errors (RMSEs) of moments of fitted distributions.

| Distribution | RMSE ($\mu_1$) | RMSE ($\mu_2$) | RMSE ($\mu_3$) | RMSE ($\mu_4$) |
|---|---|---|---|---|
| Gamma | $6.03 \times 10^{-7}$ | $2.03 \times 10^{-7}$ | $3.23 \times 10^{-8}$ | $3.60 \times 10^{-9}$ |
| Normal | $3.71 \times 10^{-6}$ | $2.62 \times 10^{-7}$ | $1.44 \times 10^{-7}$ | $1.89 \times 10^{-8}$ |
| Weibull | $1.52 \times 10^{-5}$ | $7.63 \times 10^{-7}$ | $2.41 \times 10^{-7}$ | $9.40 \times 10^{-9}$ |

In Figure 3, the NCRB of the proposed NJPE was compared with the NMSE of the simulated NJPE according to bit-energy-to-noise-power spectral-density ratio ($E_b/N_0$) with $N = 50$, and according to $N$ with $E_b/N_0 = 12$ dB. In Figure 3a, for a large SNR region, NMSE and NCRB converged to 0.04, which was also calculated by $2/N$. In Figure 3b, NMSE and NCRB were inversely proportional to the number of symbols. These results were well-matched with Lemma 1.

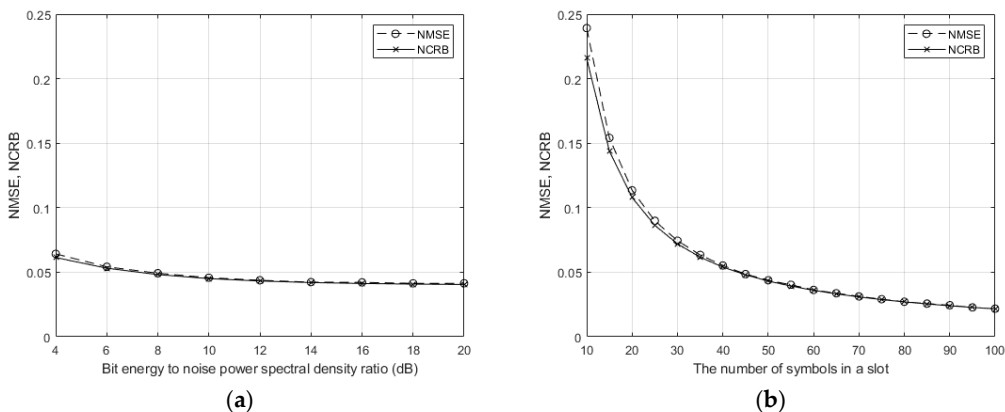

(a)  (b)

**Figure 3.** Cramér–Rao bound according to (**a**) bit-energy-to-noise-power spectral-density ratio and (**b**) number of symbols in a slot.

In Figure 4, the detection probabilities of the proposed jammer detector based on NJPE versus bit-energy-to-jammer-power spectral-density ratio ($E_b/J_0$) at $P_{fa} = 10^{-10}$ are shown for $\rho = 0.0005$, 0.005, and 0.05. The jammer with $\rho = 0.0005$ was a strong and narrowband signal. The red-triangle line denotes CA-CFAR, the green-asterisk line denotes SO-CFAR, the blue-hexagram line denotes DJR, and the magenta-diamond line denotes the proposed detector. The Gaussian-distribution-based detector is also depicted as a dotted-magenta-diamond line for comparison with the proposed gamma-distribution-based detector. When $E_b/J_0$ was small, i.e., large jammer power, the jammer was easily detected, and the detection probability of jammer was close to 1; the inverse was also true. For the same $E_b/J_0$, a small $\rho$ in Figure 4a demonstrates larger detection probability than a large $\rho$ in Figure 4c. The proposed method had about 15, 13, and 5 dB detection-probability gain, respectively, compared to the conventional CA-CFAR, SO-CFAR, and DJR. In addition, the proposed gamma-distribution-based detector had about 3 dB better gain than the Gaussian-distribution-based detector.

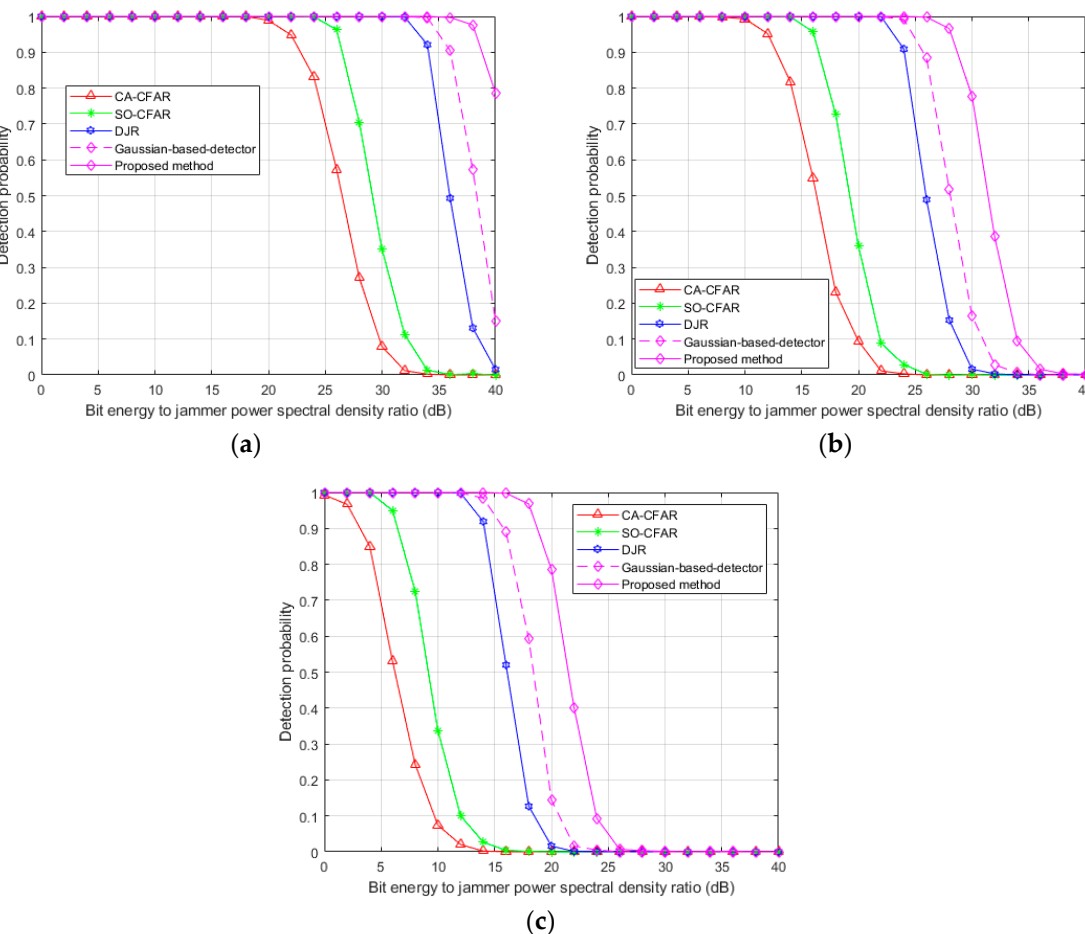

**Figure 4.** Detection probability: (**a**) $\rho = 0.0005$, (**b**) $\rho = 0.005$, (**c**) $\rho = 0.05$.

In Figure 5, the receiver-operating-characteristic (ROC) curves of the proposed method and the conventional methods were evaluated to show detection probability according to various $P_{fa}$. In Figure 5, the proposed method exhibited greater detection probability than conventional methods at small $P_{fa}$ regions, and required a smaller $P_{fa}$ for specific detection probability. As seen in Figures 4 and 5, the proposed detector demonstrated the best detection performance.

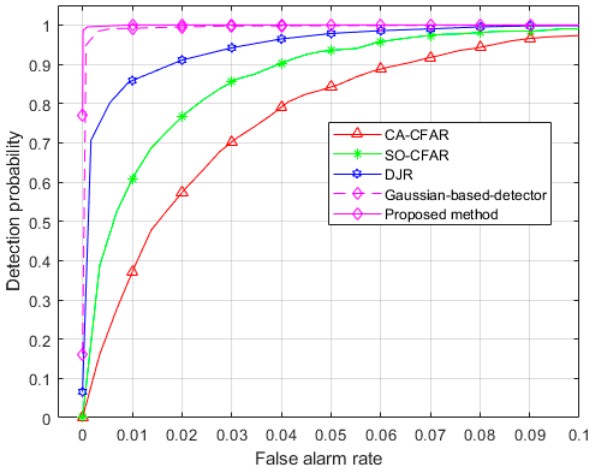

**Figure 5.** Receiver-operating-characteristic (ROC) curve for $\rho = 0.05$.

Figure 6 illustrates the BER performance of the combining slots without jammed slots versus $E_b/J_0$ at $P_{fa} = 10^{-10}$ for $\rho = 0.0005, 0.005$, and $0.05$. The tested detection methods were the same as in Figure 4. For reference comparison, the BER from combining without- and with-discard of known jammed slots is also plotted with a black solid line and a dashed line, respectively. If perfectly jammed slots were discarded, the BER was about $2 \times 10^{-4}$ regardless of $E_b/J_0$. If not, when $E_b/J_0$ was less than 5 dB, and $\rho$ was $0.0005, 0.005$, and $0.05$, the BER from combining all slots increased up to $0.001$, $0.009$, and $0.07$, respectively. These results demonstrated that rejecting jammed slots by the jammer detector was important in increasing BER performance in FHSS systems.

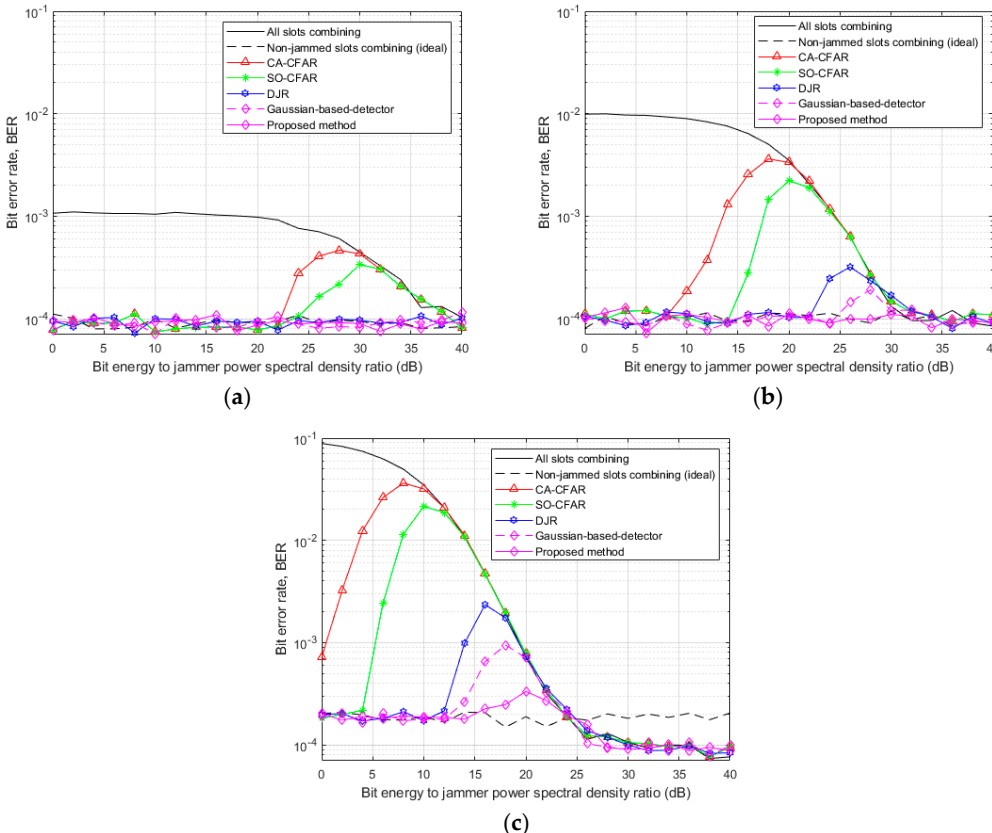

**Figure 6.** Bit-error-rate (BER) performance: (**a**) $\rho = 0.0005$, (**b**) $\rho = 0.005$, (**c**) $\rho = 0.05$.

In Figure 6, the methods ranked by increasing BER performance are CA-CFAR, SO-CFAR, DJR, and the proposed method. This result matched with detection performance in Figure 4. In Figure 6b, when $\rho$ was $0.005$ and $E_b/J_0$ varied from 10 to 30 dB, the BER performance of the proposed method was similar to that of perfect jammer detection, while the BER performance of other detectors decreased at the $E_b/J_0$ region. When $E_b/J_0$ was larger than 30 dB, the BER of all schemes encountered an error floor. In this case, even though jammer power was small, BER performance did not increase at all for all cases. As seen in BER comparisons, the proposed method demonstrated the best BER performance.

## 5. Conclusions

We proposed a NJPE and jammer detector based on gamma distribution for detecting jammed slots and combining nonjammed slots in FHSS systems. We theoretically derived the NCRB of the proposed estimator, and developed the jammer detector based on gamma distribution. By computer simulations, the derived NCRB was tested using NMSE, and BER performance for the proposed and the conventional methods was evaluated. The detection performance of the proposed method had a 15, 13, and 5 dB $E_b/J_0$ gain compared to the conventional jammer detectors. The proposed scheme showed the best detection and BER performance for all scenarios.

**Author Contributions:** Conceptualization, H.L. and J.C.; methodology, H.L.; software, H.L.; validation, J.A., Y.K., and J.C.; investigation, H.L., Y.K., and J.A.; writing—original-draft preparation, H.L.; writing—review and editing, J.C. All authors have read and agreed to the published version of the manuscript.

**Funding:** This research was supported by Inha university research program.

**Conflicts of Interest:** The authors declare no conflict of interest.

**Appendix A**

In this appendix, we calculate the parameters of the probability distribution for the proposed NJPE. Let the NJPE output be a random variable $Z$, and its sample mean and variance be $M$ and $S^2$, respectively. NJPE ($Z$) is expressed as Equation (6). We rewrite the equation as

$$Z = M - \sqrt{M^2 - S^2}. \tag{A1}$$

The multiplication of the received symbol and its complex conjugate is set as a random variable $X$. When the number of observation samples is $N$, $X_1$, $X_2$, ..., $X_N$ are assumed as independent and identically distributed (i.i.d.) random variables. Then, sample mean $M$ and sample variance $S^2$ are defined as

$$M = \frac{1}{N} \sum_{i=1}^{N} X_i, \tag{A2}$$

$$S^2 = \frac{1}{N-1} \sum_{i=1}^{N} (X_i - E[X])^2. \tag{A3}$$

The mean and variance of $M$ and $S^2$ are obtained as follows [27].

$$E[M] = \mu, \tag{A4}$$

$$\mathrm{Var}[M] = \frac{\mu_2}{N}, \tag{A5}$$

$$E[S^2] = \mu_2, \tag{A6}$$

$$\mathrm{Var}[S^2] = \frac{\mu_4}{N} - \frac{\mu_2{}^2}{N} + \frac{2\mu_2{}^2}{N(N-1)}, \tag{A7}$$

where $\mu$ and $\mu_2$ denote the mean and variance of $X$, respectively; $\mu_3$ and $\mu_4$ are the third and fourth central moment of $X$, respectively.

In FHSS, if the length of each slot was designed to be shorter than coherence time, and the channel gain was assumed to be constant in a slot. Then, random variable $X$ follows gamma distribution [28]. When the number of samples is sufficiently large, $M$ follows Gaussian distribution by the central limit theorem (CLT). Without loss of generality, the distribution of the sample variance was assumed to be Gaussian distribution [29]. As in Equation (A1), i.e., $Z = M - \sqrt{M^2 - S^2}$, since $M$ and $S^2$ follow Gaussian distributions, the distribution of $Z$ cannot be obtained in a closed form because of the square and square-root terms. This paper approximated the distribution of $Z$ as gamma distribution on the basis of simulations.

The parameters of gamma distribution were determined by the mean and variance of random variable $Z$. The mean of $Z$ could be calculated by second-order Taylor's approximation. Let $\mathrm{g}(\cdot)$ be $\sqrt{\cdot}$. Then, the mean of $Z$ is

$$\begin{aligned} E[Z] &= E[M] - E\big[\mathrm{g}\big(M^2 - S^2\big)\big] \\ &\approx E[M] - \mathrm{g}\big(E\big[M^2 - S^2\big]\big) - \frac{\mathrm{Var}[M^2 - S^2]}{2}\mathrm{g}''\big(E\big[M^2 - S^2\big]\big). \end{aligned} \tag{A8}$$

In Equation (A4), $\mathrm{E}\big[M^2 - S^2\big]$ and $\mathrm{Var}\big[M^2 - S^2\big]$ are calculated as

$$
\begin{aligned}
\mathrm{E}\big[M^2 - S^2\big] &= \mathrm{E}\big[M^2\big] - \mathrm{E}\big[S^2\big] \\
&= \mathrm{Var}[M] + \mathrm{E}[M]^2 - \mathrm{E}\big[S^2\big]
\end{aligned}
\tag{A9}
$$

$$
\mathrm{Var}\big[M^2 - S^2\big] = \mathrm{Var}\big[M^2\big] + \mathrm{Var}\big[S^2\big] - 2\mathrm{Cov}\big(M^2, S^2\big),
\tag{A10}
$$

where

$$
\begin{aligned}
\mathrm{Var}\big[M^2\big] &= \mathrm{E}\Big[\big(M^2\big)^2\Big] - \mathrm{E}\big[M^2\big]^2 \\
&= \frac{\mu_4}{N^3} + \frac{(2N-3)\mu_2{}^2}{N^3} + \frac{4\mu_3\mu}{N^2} + \frac{4\mu_2\mu^2}{N},
\end{aligned}
\tag{A11}
$$

$$
\begin{aligned}
\mathrm{Cov}\big(M^2, S^2\big) &= \mathrm{E}\big[M^2 S^2\big] - \mathrm{E}\big[M^2\big]\mathrm{E}\big[S^2\big] \\
&= \frac{\mu_4}{N(N-1)} + \frac{2\mu_3\mu}{N-1} + \mu_2\mu^2 + \mu^2.
\end{aligned}
\tag{A12}
$$

From Equations (A9) and (A10), the mean of $Z$ can be rewritten as

$$
\mathrm{E}[Z] = \mu - \sqrt{\mu^2 - \mu_2}.
\tag{A13}
$$

The variance of $Z$ can be calculated by linearizing the square root of Equation (A1). The linearized $Z$ can be expressed as

$$
Z = M - \frac{1}{2\sqrt{\mathrm{E}[M^2] - \mathrm{E}[S^2]}} M^2 + \frac{1}{2\sqrt{\mathrm{E}[M^2] - \mathrm{E}[S^2]}} S^2 - \frac{\sqrt{\mathrm{E}[M^2] - \mathrm{E}[S^2]}}{2}.
\tag{A14}
$$

The variance of $Z$ can be calculated using [27] and [30], and is written as

$$
\begin{aligned}
\mathrm{Var}[Z] &= \mathrm{Var}[M] + \left(\frac{1}{2\sqrt{\mathrm{E}[M^2] - \mathrm{E}[S^2]}}\right)^2 \times \Big(\mathrm{Var}\big[M^2\big] + \mathrm{Var}\big[S^2\big] - \mathrm{Cov}\big(M^2, S^2\big)\Big) \\
&\quad - \frac{1}{\sqrt{\mathrm{E}[M^2] - \mathrm{E}[S^2]}} \times \Big(\mathrm{Cov}\big(M, M^2\big) - \mathrm{Cov}\big(M, S^2\big)\Big) \\
&= \frac{4\mu_2\mu^2 + \mu_4 - \mu_2{}^2 - 4\mu_3\mu}{4(\mu_2 + N\mu^2 - N\mu_2)} - \frac{1}{\sqrt{\frac{\mu_2}{N} + \mu^2 - \mu_2}}\left(\frac{2\mu_2\mu}{N} - \frac{\mu_3}{N}\right) + \frac{\mu_2}{N},
\end{aligned}
\tag{A15}
$$

where

$$
\mathrm{Cov}\big(M, M^2\big) = \frac{\mu_3}{N^2} + \frac{2\mu_2\mu}{N},
\tag{A16}
$$

$$
\mathrm{Cov}\big(M, S^2\big) = \frac{\mu_3}{N}.
\tag{A17}
$$

Let the output of NJPE be $\theta$. To define mean and variance in Equations (A13) and (A15) using $\theta$, the moments of $X$ are first defined as

$$
\mu = P_r + \theta,
\tag{A18}
$$

$$
\mu_2 = 2P_r\theta + \theta^2,
\tag{A19}
$$

$$
\mu_3 = 6P_r\theta^2 + 2\theta^3,
\tag{A20}
$$

$$
\mu_4 = 12P_r{}^2\theta^2 + 36P_r\theta^3 + 9\theta^4,
\tag{A21}
$$

where $P_r$ denotes $\mathrm{E}_s\mathrm{E}\big[|h|^2\big]$. After substituting these moments of Equations (A18)–(A21) into Equations (A13) and (A15), the mean and variance are calculated as

$$
\mathrm{E}[Z] = \theta,
\tag{A22}
$$

$$\text{Var}[Z] = \frac{\theta\{(\theta + P_r)^3 + P_r(\theta - P_r)^2\}}{(N-1)P_r^2 + (\theta + P_r)^2} + \frac{\theta(\theta + 2P_r)}{N} - \frac{4P_r^2\theta}{\sqrt{N(N-1)P_r^2 + N(\theta + P_r)^2}}. \tag{A23}$$

When the number of samples is sufficiently large, Equation (A23) can be simplified as

$$\text{Var}[Z] = \frac{\theta^2(\theta^2 + 4P_r\theta + 2P_r^2)}{NP_r^2}. \tag{A24}$$

Therefore, the conditional PDF of $z$, given $\theta$, $f_Z(z|\theta)$, is given as

$$f_Z(z|\theta) = \frac{1}{\Gamma(\alpha(\theta))\beta(\theta)^{\alpha(\theta)}} z^{\alpha(\theta)-1} \exp(-z/\beta(\theta)), \tag{A25}$$

where

$$\alpha(\theta) = \frac{\text{E}[Z]^2}{\text{Var}[Z]} = \frac{NP_r^2}{\theta^2 + 4P_r\theta + 2P_r^2}, \tag{A26}$$

$$\beta(\theta) = \frac{\text{Var}[Z]}{\text{E}[Z]} = \frac{\theta(\theta^2 + 4P_r\theta + 2P_r^2)}{NP_r^2}. \tag{A27}$$

## Appendix B

In this appendix, the CRB of the proposed NJPE using Fisher information is derived. CRB is defined as $\text{Var}[\hat{\theta}] \geq 1/I(\theta)$ [24], and $I(\theta)$ is defined as

$$I(\theta) = \text{E}\left[\left(\frac{\partial \ln f_Z(z|\theta)}{\partial \theta}\right)^2\right] = -\text{E}\left[\frac{\partial^2 \ln f_Z(z|\theta)}{\partial \theta^2}\right], \tag{A28}$$

where $\ln f_Z(z|\theta)$ can be written as

$$\begin{aligned}
\ln f_Z(z|\theta) &= \ln\left(\frac{1}{\Gamma(\alpha(\theta))\beta(\theta)^{\alpha(\theta)}} z^{\alpha(\theta)-1} \exp(-z/\beta(\theta))\right) \\
&= -\ln\Gamma(\alpha(\theta)) - \alpha(\theta)\ln\beta(\theta) + (\alpha(\theta)-1)\ln z - \frac{z}{\beta(\theta)}.
\end{aligned} \tag{A29}$$

From [31], Equation (A29) can be rewritten as

$$\begin{aligned}
\ln f_Z(z|\theta) = {}& -\alpha(\theta)\ln\alpha(\theta) + \tfrac{1}{2}\ln\alpha(\theta) + \alpha(\theta) - \tfrac{1}{2}\ln 2\pi \\
& -\alpha(\theta)\ln\beta(\theta) + (\alpha(\theta)-1)\ln z - \frac{z}{\beta(\theta)}.
\end{aligned} \tag{A30}$$

The second derivative of $\theta$ in Equation (A30) is given as

$$\begin{aligned}
\frac{\partial^2 \ln f_Z(z|\theta)}{\partial \theta^2} = {}& \frac{\partial^2\alpha(\theta)}{\partial\theta^2}\left(\ln z - \ln\theta + \frac{1}{2\alpha(\theta)}\right) - \frac{\partial\alpha(\theta)}{\partial\theta}\left(\frac{1}{\theta} + \frac{\partial\alpha(\theta)}{\partial\theta}\frac{1}{2\alpha(\theta)^2}\right) \\
& + \left(\frac{\partial^2\beta(\theta)}{\partial\theta^2}\frac{1}{\beta(\theta)^2} - \left(\frac{\partial\beta(\theta)}{\partial\theta}\right)^2\frac{2}{\beta(\theta)^3}\right)(z-\theta) - \frac{\partial\beta(\theta)}{\partial\theta}\frac{1}{\beta(\theta)^2}.
\end{aligned} \tag{A31}$$

From Equations (A28) and (A31), $I(\theta)$ can be calculated as

$$\begin{aligned}
I(\theta) = {}& -\text{E}\left[\frac{\partial^2 \ln f_Z(z|\theta)}{\partial\theta^2}\right] \\
= {}& -\frac{\partial^2\alpha(\theta)}{\partial\theta^2}\left(\text{E}[\ln Z] - \ln\theta + \frac{1}{2\alpha(\theta)}\right) + \frac{\partial\alpha(\theta)}{\partial\theta}\left(\frac{1}{\theta} + \frac{\partial\alpha(\theta)}{\partial\theta}\frac{1}{2\alpha(\theta)^2}\right) \\
& -\left(\frac{\partial^2\beta(\theta)}{\partial\theta^2}\frac{1}{\beta(\theta)^2} - \left(\frac{\partial\beta(\theta)}{\partial\theta}\right)^2\frac{2}{\beta(\theta)^3}\right)(\text{E}[Z] - \theta) + \frac{\partial\beta(\theta)}{\partial\theta}\frac{1}{\beta(\theta)^2},
\end{aligned} \tag{A32}$$

where $\mathrm{E}[Z]$ is equal to $\theta$ by Equation (A22), and $\mathrm{E}[\ln Z]$ is calculated using second-order Taylor's approximation, and calculated as

$$\mathrm{E}[\ln Z] = \ln \mathrm{E}[Z] - \frac{\mathrm{Var}[Z]}{2\mathrm{E}[Z]^2} = \ln \theta - \frac{1}{2\alpha(\theta)}. \tag{A33}$$

Using Equation (A33), Equation (A32) can be rewritten as

$$I(\theta) = \frac{\partial \alpha(\theta)}{\partial \theta}\left(\frac{1}{\theta} + \frac{\partial \alpha(\theta)}{\partial \theta}\frac{1}{2\alpha(\theta)^2}\right) + \frac{\partial \beta(\theta)}{\partial \theta}\frac{1}{\beta(\theta)^2}. \tag{A34}$$

Fisher information $I(\theta)$ is calculated as

$$I(\theta) = \frac{NP_r{}^2\left(\theta^2 + 4P_r\theta + 2P_r{}^2\right) + 2\theta^2(\theta + P_r)^2}{\theta^2(\theta^2 + 4P_r\theta + 2P_r{}^2)^2}. \tag{A35}$$

When the number of samples is sufficiently large, Equation (A35) is simplified as

$$I(\theta) = \frac{NP_r{}^2}{\theta^2(\theta^2 + 4P_r\theta + 2P_r{}^2)}. \tag{A36}$$

Therefore, CRB is obtained as

$$\mathrm{Var}\!\left[\hat{\theta}\right] \geq CRB(\theta) = \frac{\theta^2\left(\theta^2 + 4P_r\theta + 2P_r{}^2\right)}{NP_r{}^2}. \tag{A37}$$

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
