# Peer review of "Antijamming Improvement for Frequency Hopping Using Noise-Jammer Power Estimator"

_applsci, doi:10.3390/app10051733_

Round 1

Reviewer 1 Report

The presentation of the proposed technique is mathematically sound and supported by the simulation results. The appendix is a bit lengthy but I find it necessary for this kind of analysis.

In any case, I have no remarks apart from the need for a minor editing regarding the use of english.

The article is certainly publishable.

Author Response

Thanks for your comments.

Reviewer 2 Report

The paper presents an analysis and validation of the performance of a jamming detector based on the Gamma distribution assumption. The explanation of the method is clear and the paper might be improved by addressing the following issues:

  • a more exhaustive validation of the assumption of the Gamma distribution for the variable Z; some exemplary curves of the real and approximate distributions could be presented clarifying if there is a specific range of the parameters in which the approximation is valid or not. Also the assumption on the jamming signal - Gaussian as the AWGN - should be discussed w.r.t. the validity of the method since jamming signal could be for example also strong narrowband signals, really different from AWGN.
  • The numerical results should be completed with other simulations / analysis, in particular w.r.t. two parameters: \rho, the ratio between the bandwidths, which is limited to 2 low values and the false alarm probability, limited to one extremely low value, almost unrealistic in practical applications. This point is also related to the previous one and it would be important to have curves of the detection probability vs. false alarm and performance in presence of different jamming signals or at least with a range of different bandwidths.

Author Response

Thanks for the review.

Round 2

Reviewer 2 Report

In the revised version, the authors have responded to the issues, especially those related to the numerical results and statistical models.